# Utility of Contrast-Enhanced Harmonic Endoscopic Ultrasonography for Early Diagnosis of Small Pancreatic Cancer

**DOI:** 10.3390/diagnostics10010023

**Published:** 2020-01-01

**Authors:** Yasunobu Yamashita, Kensuke Tanioka, Yuki Kawaji, Takashi Tamura, Junya Nuta, Keiichi Hatamaru, Masahiro Itonaga, Takeichi Yoshida, Yoshiyuki Ida, Takao Maekita, Mikitaka Iguchi, Masaki Terada, Tetsuo Sonomura, Seiko Hirono, Ken-Ichi Okada, Manabu Kawai, Hiroki Yamaue, Masayuki Kitano

**Affiliations:** 1Second Department of Internal Medicine, Wakayama Medical University, Wakayama 641-8509, Japan; y.kawaji1985@gmail.com (Y.K.); ttakashi@wakayama-med.ac.jp (T.T.); jnuta721@wakayama-med.ac.jp (J.N.); papepo51@wakayama-med.ac.jp (K.H.); masaitonaga0907@gmail.com (M.I.); tayoshid@wakayama-med.ac.jp (T.Y.); y-mori@wakayama-med.ac.jp (Y.I.); maekita@wakayama-med.ac.jp (T.M.); mikitaka@wakayama-med.ac.jp (M.I.); kitano@wakayama-med.ac.jp (M.K.); 2Clinical Study Support Center, Wakayama Medical University Hospital, Wakayama 641-8509, Japan; ktaniok@wakayama-med.ac.jp; 3Department of Radiology, Wakayama Minami Radiology Clinic, Wakayama 641-0012, Japan; m-terada@pet-wakayama.com; 4Department of Radiology, Wakayama Medical University, Wakayama 641-8509, Japan; sonomura@wakayama-med.ac.jp; 5Second Department of Surgery, Wakayama Medical University, Wakayama 641-8509, Japan; seiko-h@wakayama-med.ac.jp (S.H.); okada@wakayama-med.ac.jp (K.-I.O.); kawai@wakayama-med.ac.jp (M.K.); yamaue-h@wakayama-med.ac.jp (H.Y.)

**Keywords:** early diagnosis of pancreatic cancer, contrast-enhanced harmonic endoscopic ultrasonography, magnetic resonance imaging, multidetector computed tomography

## Abstract

This study aimed to assess whether contrast-enhanced harmonic endoscopic ultrasonography (CH-EUS), compared to multidetector-row computed tomography (MDCT) and magnetic resonance imaging (MRI), is useful for early diagnosis of small pancreatic cancer (PC). Between March 2010 and June 2018, all three imaging modalities and surgery were performed for patients with a pancreatic solid lesion measuring ≤20 mm; diagnostic ability was compared among modalities. Fifty-one of 60 patients were diagnosed with PC (PC size in 41 patients: 11–20 mm; 10 patients: ≤10 mm). The sensitivity, specificity, and accuracy of CH-EUS, MDCT, and MRI for PC (11–20 mm) detection were 95%/83%/94%, 78%/83%/79%, and 73%/33%/68%, respectively. The diagnostic ability of CH-EUS was significantly superior compared with MDCT and MRI (*p* = 0.002 and *p* = 0.007, respectively). The sensitivity, specificity, and accuracy of CH-EUS, MDCT, and MRI for PC (≤10 mm) detection were 70%/100%/77%, 20%/100%/38%, and 50%/100%/62%, respectively. The diagnostic ability of CH-EUS tended to be superior to that of MDCT (*p* = 0.025). The sensitivity of MDCT for PC (≤10 mm) detection was significantly lower than that for PC (11–20 mm) detection (20% vs. 78%; *p* = 0.001). CH-EUS, compared to MDCT and MRI, is useful for diagnosing small PCs.

## 1. Introduction

Currently, the incidence and mortality rate of pancreatic cancer are rising rapidly, and it is the fourth leading cause of cancer death in the USA and Japan [1,2]. Even with recent advancements in diagnostic imaging modalities, most cases of pancreatic cancer are discovered at an unresectable stage when the prognosis is poor and the 5-year survival rate is 3% [1]. 

However, Agarwal et al. reported that the early diagnosis of pancreatic cancer should be achieved before the lesion reaches 20 mm in size, as a major shift in surgical resectability and patient survival occurs when the lesions increase in size from 20 to 30 mm. The median survival time of pancreatic cancer measuring ≤20 mm was 17.2 ± 8.2 months [3]. Moreover, according to the Japan Pancreatic Cancer Registry (JPCR), the 5-year survival rate of patients with lesions smaller than 1 cm and between 1–2 cm in diameter were 80.4% and 50.4%, respectively [4]. Therefore, we should diagnose small pancreatic cancer lesions measuring ≤20 mm for early surgical resection. Moreover, diagnosing these lesions while they are ≤10 mm in size leads to a better prognosis. However, only 0.8% patients with lesions smaller than 10 mm have been registered in JPCR [4]. 

For the diagnosis of pancreatic cancer, multidetector-row computed tomography (MDCT) and magnetic resonance imaging (MRI) are the two most commonly used imaging modalities. Endoscopic ultrasonography (EUS) is thought to be one of the most reliable and efficient diagnostic modalities for pancreatobiliary diseases because it is superior to any other modality with respect to spatial resolution [5,6]. Despite its detection capabilities, conventional EUS has a limitation in characterizing pancreatic lesions because most pancreatic solid lesions appear as hypoechoic lesions. Therefore, contrast-enhanced harmonic EUS (CH-EUS) is one way to improve the ability to characterize pancreatic lesions with evaluation of vascularity. In fact, Kitano et al. reported a depiction of hypoenhancement on CH-EUS diagnosed pancreatic cancer with a sensitivity (95%) and specificity (89%) on the largest series of 277 subjects [7]. This data demonstrated the usefulness and high diagnostic ability of CH-EUS in diagnosing pancreatic cancer.

Therefore, in this study, we evaluated the diagnostic ability of CH-EUS for pancreatic cancer, in comparison to MDCT and MRI, as they have been commonly used in the clinical management of small pancreatic cancers.

## 2. Materials and Methods

### 2.1. Patients

Patients with a pancreatic solid lesion measuring ≤20 mm who underwent surgery after undergoing evaluation with all three image modalities (CH-EUS, MDCT, and MRI) between March 2010 and June 2018 were enrolled in this study. Only surgical cases were included in this study for accurate tumor size determination based on histological measurement of specimens. Therefore, the final diagnosis was based on the histopathology of the surgical specimens in all cases. The lesion size was calculated based on measurements of the surgical specimen. Patients were divided into two groups according to the final diagnosis: pancreatic cancer and non-pancreatic cancer. The inclusion criteria were as follows: (1) patient age older than 20 years; (2) pancreatic solid lesion measuring ≤20 mm; (3) all three imaging modalities were performed; (4) pathological diagnosis with resected specimens was obtained; and (5) surgery was performed on pancreatic solid lesions. The exclusion criteria were as follows: (1) inability to carry out any of the imaging modalities and (2) inability to perform a pancreas screening with EUS due to surgical altered anatomy. The patients were divided into two groups based on lesion size (11–20 mm and ≤10 mm) for analysis. This study was approved by the ethics committee of Wakayama Medical University (No. 2458). Informed consent was obtained from each patient.

### 2.2. Study Design

This study was a retrospective observational study performed at Wakayama Medical University Hospital. The primary outcome measurement was to evaluate the diagnostic ability of CH-EUS versus MDCT and MRI in diagnosing small pancreatic cancers.

### 2.3. EUS Procedure 

EUS (GF-UE260-AL5, GF-UCT260; Olympus, Tokyo, Japan) with ultrasound observation systems (ALOKA ProSound SSD α-10; Aloka Co. Ltd, Tokyo, Japan) was performed under sedation. Fundamental B-mode EUS was initially performed. After using the fundamental B-mode, CH-EUS was performed for the differential diagnosis of the pancreatic lesion. CH-EUS was performed with the extended pure harmonic detection method for which the mechanical index was set at 0.35. Sonazoid (Daiichi Sankyo Co. Ltd., Tokyo, Japan) was used to perform CH-EUS. It is a second-generation ultrasonography contrast agent composed of perfluorobuthane microbubbles whose median diameter was 2–3 μm [8]. Vascular patterns were continuously assessed in real-time after infusion of contrast agent. Two endosonographers evaluated EUS images When there was a conflicting assessment between the two reviewers, they re-evaluated the data together until an agreement was reached. First, they evaluated whether a pancreatic nodule was present or not. Second, when a nodule was present, they evaluated whether the nodule was hypovascular compared to the surrounding tissue. Pancreatic cancer was defined as a hypovascular nodule on CH-EUS. 

### 2.4. MDCT

MDCT scans were performed initially without contrast on a 64 or 320 slice scanner (Aquilion; Canon Medical Systems Corporation, Tochigi, Japan). Two-millimeter sections were acquired in the peripancreatic area. Then, 1.6 mL/kg nonionic contrast material with an iodine concentration of 300 mgI/mL was injected with a fixed duration of 30 s. The arterial phase was initiated after the bolus tracking program detected the threshold enhancement of 200 HU in the aorta using an automatic bolus-tracking method. After administration, the pancreatic parenchymal, portal, and delayed phases were fixed at 15, 30, and 120 s, respectively. MDCT images were analyzed separately and independently by two expert radiologists. First, they evaluated whether a pancreatic nodule was present or not. Second, when nodule was present, they evaluated whether the nodule was hypovascular compared with the surrounding tissue. The pancreatic cancer was defined as a hypovascular nodule on MDCT. When there was a difference in assessment between the two reviewers, they re-evaluated the data together until an agreement was reached.

### 2.5. MRI

MRI was performed using a 3.0 Tesla machine (Achieva 3.0 T; Philips, Tokyo, Japan). Diagnosis was performed with coronal balanced steady-state free precession imaging (4 mm slices), coronal and axial T2-weighted single-shot fast spin-echo series (4 mm slices), fat-suppressed T2-weighted fast spin-echo series (3 mm to 3.5 mm slices), 3D heavily T2-weighted coronal MR cholangiopancreatography (1.8 mm slices), and axial diffusion-weighted imaging, including apparent diffusion coefficient (ADC) mapping (4 mm slices) using three b-values (b = 0, 50, and 1000 s/mm^2^). Two radiologists who were blinded to the results read the MR images. First, they evaluated whether pancreatic nodule was present or not. Second, when a nodule was present on MRI, they evaluated the nodule based on diffusion-weighted imaging. When a nodule had an increased signal intensity on diffusion-weighted images, with high b values and relatively low ADC values, the lesion was diagnosed as pancreatic cancer. When there was a difference in assessment between the two reviewers, they re-evaluated the data together until an agreement was reached.

### 2.6. Statistical Analysis

In terms of comparison for diagnostic ability among imaging modalities, the *p* value < 0.017 was considered to denote statistical significance according to Bonferroni correction for 0.05 decision threshold. Fisher’s exact test was used compare diagnostic ability of cancer based on size. A difference was considered significant when the *p* value was less than 0.05. Statistical analysis was performed with JMP Pro version 13 (SAS Institute Inc., Cary, NC, USA).

## 3. Results

A total of 60 patients were enrolled in the study. The final pathological diagnoses of the solid pancreatic lesions in the patients were as follows: pancreatic cancer (n = 51), neuroendocrine tumor (NETs; n = 5), solid pseudopapillary neoplasm (SPN; n = 1), and chronic pancreatitis (CPs; n = 3).

There were 47 patients with pancreatic lesions ranging between 11 mm and 20 mm in size. Forty-one patients (22 males, 19 females; median age 70 years, range 43–88 years) had pancreatic cancer. The lesion size was 17.1 ± 2.6 mm (mean ± SD). Seventeen of 41 patients (41%) had no symptoms, and 26 of 41 patients (63%) exhibited pancreatic duct dilation (Table 1). Six had benign lesions (NET, *n* = 2; CP, *n* = 3; SPN, *n* = 1). The sensitivity, specificity, and accuracy for pancreatic cancer ranging between 11–20 mm in size with CH-EUS, MDCT, and MRI were 95%/83%/94%, 78%/83%/79%, and 73%/33%/68%, respectively. CH-EUS was significantly superior to MDCT and MRI with respect to diagnosing pancreatic cancer (*p* = 0.02 and *p* = 0.007, respectively) (Table 2 and Table 3).

There were 13 patients with pancreatic lesions ≤10 mm in size. Ten patients (4 males, 6 females; median age 73 years, range 44–84 years) had pancreatic cancer, including three cases of carcinoma in situ. The lesion size was 5.1 ± 3.9 mm (mean ± SD). Nine of 10 patients (90%) with pancreatic cancer had no symptoms or pancreatic duct dilation (Table 4). Three patients had benign lesions (NET, *n* = 3). The sensitivity, specificity, and accuracy for pancreatic cancer measuring ≤10 mm with CH-EUS, MDCT, and MRI were 70%/100%/77%, 20%/100%/39%, and 50%/100%/62%, respectively. The diagnostic ability of CH-EUS tended to be superior to MDCT in diagnostic ability (*p* = 0.025). (Table 5 and Table 6) (Figure 1). In MDCT, the sensitivity for pancreatic cancer measuring ≤10 mm (20%) was significantly lower than that for pancreatic cancer measuring 11–20 mm (78%) (*p =* 0.001). There was no significant difference in CH-EUS and MRI with regard to the diagnostic ability of cancer based on size.

## 4. Discussion

Pancreatic cancer has a dismal prognosis, with a 5-year overall survival rate of 8% [2]. Therefore, early diagnosis is necessary and remains the only hope for improved prognosis in patients with pancreatic cancer. Thus, it is also necessary to identify an effective diagnostic imaging modality for early pancreatic cancer.

In this study, we evaluated the diagnostic ability of three imaging modalities (CH-EUS, MDCT, and MRI) to diagnosis small pancreatic cancer according to size (11–20 mm versus ≤10 mm). With regard to diagnostic ability, CH-EUS was significantly superior to MDCT and MRI in diagnosing pancreatic cancer measuring 11–20 mm. CH-EUS tended to be superior to MDCT for diagnosing pancreatic cancer measuring ≤10 mm. With MDCT, the sensitivity was significantly lower for lesions measuring ≤10 mm (20%) than for those measuring 11–20 mm (78%) (Table 2 and Table 5). 

CH-EUS demonstrated good performance in diagnostic ability compared with the other two imaging modalities (MDCT and MRI). The following are potential reasons for these results: (1) EUS is superior to any other modality with respect to spatial resolution; and (2) EUS allows for dynamic and repeated examinations in an area suspected be pancreatic cancer. However, CH-EUS is superior to conventional EUS, and its utility for the differential diagnosis of pancreatic lesions has been reported in many studies [7,9,10,11,12,13,14,15,16,17,18,19,20,21,22,23,24,25], as most pancreatic solid lesions are depicted as a hypoechoic mass in conventional EUS regardless of histology. Therefore, we used CH-EUS in the differential diagnosis for pancreatic lesions. 

There were five previous studies [7,26,27,28,29] that reported the utility of EUS versus MDCT in diagnosis of pancreatic cancer measuring ≤20 mm. Three studies included less than 10 cases each [26,27,28], and one study included less than 20 cases each [29]. Statistical analyses were not performed in three studies enrolling less than ten cases due to the small number of patients included [26,27,28]. There was only one study reporting the utility of CH-EUS for small pancreatic cancers, which found that CH-EUS was significantly superior to MDCT for the differential diagnosis of solid pancreatic lesions [7]. This study finding was consistent with our results. Our study has an advantage over other previous reports. First, most previous reports comparing EUS and MDCT for small pancreatic cancers enrolled a small number of patients (less than 20 patients). Second, there was no study that compared multiple modalities, including MDCT and MRI. Our study demonstrated that CH-EUS was significantly superior to MDCT and MRI. Third, there are currently no studies comparing the ability of imaging modalities to diagnosis pancreatic cancer measuring less than 10 mm. Our study demonstrated CH-EUS was significantly superior to MDCT for pancreatic cancer measuring ≤10 mm. Additionally, our study highlighted that the sensitivity of MDCT for pancreatic cancer measuring ≤10 mm was significantly lower than that for pancreatic cancer measuring 11–20 mm. To our knowledge, this is the first report demonstrating the superiority of CH-EUS to MDCT and MRI in the diagnosing of small pancreatic cancers. 

With regard to patient characteristics, although 90% of patients with a lesion measuring ≤10 mm had no symptoms in our study, 9 of 10 patients (90%) exhibited pancreatic duct dilation. For the early diagnosis of small pancreatic cancer, we should perform CH-EUS in those with pancreatic duct dilatation and no symptoms, even if the pancreatic lesion is not detected by MDCT and MRI.

Our study has several limitations. First, this study included a small number of cases of pancreatic cancer measuring ≤10 mm. Second, approximately 80% of the enrolled patients were pancreatic cancers.

## 5. Conclusions

Although MDCT and MRI are often the first-line imaging modalities used for the diagnosis of suspected pancreatic cancer lesions, CH-EUS is significantly superior to MDCT and MRI in terms of diagnosing small pancreatic cancer. Thus, we strongly recommend CH-EUS as a method for the early diagnosis of small pancreatic cancer. Moreover, patients with imaging findings indicating dilation of the main pancreatic duct should undergo screening with EUS. If EUS reveals a pancreatic lesion, CH-EUS should be considered for the differential diagnosis.

## Figures and Tables

**Figure 1 diagnostics-10-00023-f001:**
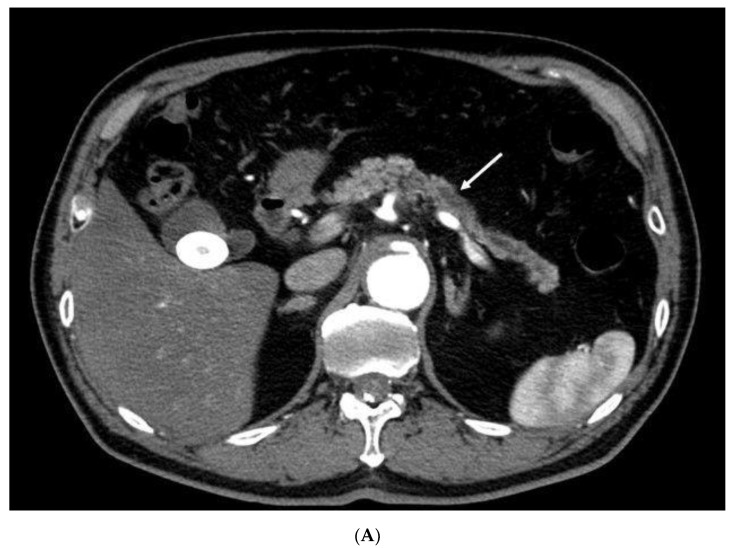
Contrast-enhanced endoscopic ultrasonography detection for small pancreatic cancer missed on contrast-enhanced multidetector-row computed tomography. (**A**) Contrast-enhanced multidetector-row computed tomography (MDCT). Although main pancreatic duct dilation (arrow) is depicted with contrast-enhanced MDCT, pancreatic lesion is not detected. (**B**) Contrast-enhanced harmonic endoscopic ultrasonography (CH-EUS). A hypoechoic lesion of 8 mm in size (arrowheads), with main pancreatic duct dilation (arrow), is demonstrated in the pancreas with fundamental B-mode EUS (left). CH-EUS (right) shows that the lesion (arrowheads) with main pancreatic duct dilation (arrow) has echo signal of lower intensity than that of surrounding pancreatic tissue.

**Table 1 diagnostics-10-00023-t001:** Characteristics of patients with pancreatic cancer measuring 11–20 mm.

Total Number of Patients	41
Sex (male/female)	22/19
Age, years (range)	70 (43–88)
Lesion size, mm (mean ± SD)	17.1 ± 2.6
Lesion location (head/body-tail)	28/13
Symptoms (present/absent)	24/17
Pancreatic duct dilation (present/absent)	26/15

**Table 2 diagnostics-10-00023-t002:** Diagnostic ability of three modalities (CH-EUS, MDCT, MRI) for pancreatic lesion measuring 11–20 mm.

	Lesion Size (11–20 mm) (*n* = 47)			
	Sensitivity	Specificity	Accuracy	*p*-Value
CH-EUS vs. MDCT	95% vs. 78%	83% vs. 83%	94% vs. 79%	0.02
CH-EUS vs. MRI	95% vs. 73%	83% vs. 33%	94% vs. 68%	0.007
MDCT vs. MRI	78% vs. 73%	83% vs. 33%	79% vs. 68%	0.41

**Table 3 diagnostics-10-00023-t003:** Diagnostic ability of CH-EUS for pancreatic lesions measuring 11–20 mm.

CH-EUS Findings	Final Diagnosis (Pathology)	
	Pancreatic Cancer (*n* = 41)	Non-Pancreatic Cancer (*n* = 6)
Pancreatic cancer	39	1
Non-pancreatic cancer	2	5

CH-EUS, contrast-enhanced harmonic endoscopic ultrasonography; MDCT, multidetector-row computed tomography; MRI, magnetic resonance imaging.

**Table 4 diagnostics-10-00023-t004:** Characteristics of patients with pancreatic cancer measuring ≤10 mm.

Total Number of Patients	10
Sex (male/female)	4/6
Age, years (range)	73 (44–84)
Lesion size, mm (mean ± SD)	5.1 ± 3.9
Lesion location (head/body-tail)	4/6
Symptoms (present/absent)	1/9
Pancreatic duct dilation (present/absent)	9/1

**Table 5 diagnostics-10-00023-t005:** Diagnostic ability of three modalities (CH-EUS, MDCT, MRI) for pancreatic lesion measuring ≤10 mm.

	Lesion Size ≤10 mm (*n* = 13)			
	Sensitivity	Specificity	Accuracy	*p*-Value
CH-EUS vs. MDCT	70% vs. 20%	100% vs. 100%	77% vs. 39%	0.025
CH-EUS vs. MRI	70% vs. 50%	100% vs. 100%	77% vs. 62%	0.16
MDCT vs. MRI	20% vs. 50%	100% vs. 100%	39% vs. 62%	0.08

**Table 6 diagnostics-10-00023-t006:** Diagnostic ability of CH-EUS for pancreatic lesions measuring ≤10 mm.

CH-EUS Findings	Final Diagnosis (Pathology)	
	Pancreatic Cancer (*n* = 10)	Non-Pancreatic Cancer (*n* = 3)
Pancreatic cancer	7	0
Non-pancreatic cancer	3	3

CH-EUS, contrast-enhanced harmonic endoscopic ultrasonography; MDCT, multidetector-row computed tomography; MRI, magnetic resonance imaging.

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
