# Peer review of "Utility of Contrast-Enhanced Harmonic Endoscopic Ultrasonography for Early Diagnosis of Small Pancreatic Cancer"

_diagnostics, 2020, doi:10.3390/diagnostics10010023_

Round 1

Reviewer 1 Report

Pancreatic cancer is one of the deadliest cancer with highest 5 year post diagnostic mortality. Surgery is currently the most effective treatment option but less than 20% of the patients are eligible due to late diagnosis. Thus, correct diagnosis of early pancreatic lesions can significantly improve patient outcome. In this article authors report a retrospective study comparing the diagnostic ability of CH-EUS, MDCT and MRI imaging techniques for the detection of early stage pancreatic cancer. Authors found CH-EUS to be more effective in detecting small pancreatic cancer lesions compared to MCDT or MRI. The paper is well written, clear and concise. Minor English (sentence construction) and text editing will be helpful. For example, citation in the text is not consistent throughout the text (Discussion section, line 1).

Author Response

We appreciate the reviewer's constructive comments.

Authrs' reply: We apologize for the error in reference numbers. The reference numbers have been corrected in the revised manuscript.

Reviewer 2 Report

In current manuscript, Yamashita et al. compared the diagnostic ability of contrast-enhanced harmonic endoscopic ultrasonography (CH-EUS), MRI, and MDCT in the early diagnosis of small pancreatic cancer (PC). The results have shown that CH-EUS has higher sensitivity, specificity, and accuracy in the detection of pancreatic lesions with a size between 11-20mm. Despite the relatively small number of patients. the study was properly designed and performed, and can potentially provide useful guidance for the early diagnosis of PC. There are some minor issues that need to be addressed:

1) The introduction part should include a few sentences to briefly describe the difference CH-EUS with conventional EUS. Such information is important for readers to understand the rationale of the study.

2) Some of the previous studies on the usage of CH-EUS in the diagnosis of pancreatic diseases should be cited. For example, Michiko et al, 2009, Ohno et al. 2012.

Author Response

We appreciate the reviewer's constructive  comments.

Minor 1

Authors' reply: In previous manuscript, this point had been included in the introduction section " conventional EUS has a limitation in characterizing pancreatic lesions because most pancreatic solid lesions appear as hypoechoic lesions. Therefore, CH-EUS is one way to improve the ability to characterize pancreatic lesions with evalustion of vascularity"

Minor 2

Authors' reply: We agree the reviewer on this point. We added usages of CH-EUS in diagnosis of pancreatic cancer with previous report. 

Reviewer 3 Report

The authors described the usefulness of CH-EUS analysis for the diagnosis of pancreatic tumors. The reviewers also think CH-EUS analysis was a usefulness method for the pancreatic diseases. However, there are several problems in this study.

 Major comments

1. In this study, included patients were only the patients whose diagnosis were with resected specimens. Why do you include the patients with EUS-FNA?

2. It is unclear what the diagnostic ability is in this study. I think authors divide the patients into two groups (pancreatic cancer, and other diseases). Is it true? If so, please describe in the methods.

3. In this study, only diagnostic ability (accuracy, sensitivity, and specificity) were described in result. If you can, Cross tabulation (2*2) tables is required because specificity of 11-20mm cohorts was lower than that of <10mm cohorts.

4. In statistical analysis session, you use McNemar test. I think this analysis is multiple comparison (CT vs MRI, CT vs EUS, and EUS vs MRI), and familywise error rate is existed. You need to correct methods such as Bonferroni correction and so on.

Author Response

We appreciate the reviewer's careful and critical comments.

Major 1

Authors' reply: Only surgical cases were included for accurate tumor size measurement with specimens. We have added this reason in methods section.

Major 2

Authors' reply: Thank you for your comment. We modified the unclear parts in the methd section as required.

Major 3

Authors' reply: We have added tables according to the reviewer's comment.

Major 4

We agree with the reviewer on this point. Therefore, we analyzed data with the correct method (Bonferroni correction).

Round 2

Reviewer 3 Report

The authors revised the enough  manuscript required by the reviewers